# Lethal Dose Conjecture on Data Poisoning

**Wenxiao Wang, Alexander Levine and Soheil Feizi**
Department of Computer Science
University of Maryland
College Park, MD 20742
`{wwx, alevine0, sfeizi}@umd.edu`

## Abstract

Data poisoning considers an adversary that distorts the training set of machine learning algorithms for malicious purposes. In this work, we bring to light one conjecture regarding the fundamentals of data poisoning, which we call the **Lethal Dose Conjecture**. The conjecture states: If $n$ clean training samples are needed for accurate predictions, then in a size-$N$ training set, only $\Theta(N/n)$ poisoned samples can be tolerated while ensuring accuracy. Theoretically, we verify this conjecture in multiple cases. We also offer a more general perspective of this conjecture through distribution discrimination. Deep Partition Aggregation (DPA) and its extension, Finite Aggregation (FA) are recent approaches for provable defenses against data poisoning, where they predict through the majority vote of many base models trained from different subsets of training set using a given learner. The conjecture implies that both DPA and FA are (asymptotically) optimal—if we have the most data-efficient learner, they can turn it into **one of the most robust defenses** against data poisoning. This outlines a practical approach to developing stronger defenses against poisoning via finding data-efficient learners. Empirically, as a proof of concept, we show that by simply using different data augmentations for base learners, we can respectively **double** and **triple** the certified robustness of DPA on CIFAR-10 and GTSRB without sacrificing accuracy.

## 1  Introduction

With the increasing popularity of machine learning and especially deep learning, concerns about the reliability of training data have also increased: typically, because of the availability of data, many training samples have to be collected from users, internet websites, or other potentially malicious sources for satisfying utilities. This motivates the development of the data poisoning threat model, which focuses on the reliability of models trained from adversarially distorted data [13].

Data poisoning is a class of training-time adversarial attacks: The adversary is given the ability to insert and remove a bounded number of training samples to manipulate the behavior (e.g. predictions for some target samples) of models trained using the resulted, poisoned dataset. Models trained from the poisoned dataset are referred to as poisoned models and the allowed number of insertion and removal is the attack size.

To challenge the reliability of poisoned models, many variants of poisoning attacks have been proposed, including triggerless attacks [28, 37, 1, 12], which do not modify target samples, and backdoor attacks [5, 32, 27], which modify them. In addition, in cases where the adversary can achieve its goal with a non-negligible probability by doing nothing, a theoretical study [23] discovers a provable attack replacing $\tilde{O}(\sqrt{N})$ samples in a size-$N$ training set. Meanwhile, defenses against data poisoning are also emerging, including detection-based defenses [30, 10, 25, 31, 33], trying to identify poisoned samples, and training-based defenses [35, 26, 22, 15, 20, 17, 4, 34], aiming at robustifying models trained from poisoned data.

36th Conference on Neural Information Processing Systems (NeurIPS 2022).

In this work, we target the fundamentals of data poisoning and ask:

- **What amount of poisoned samples will make a *specific* task impossible?**

The answer we propose is the following conjecture, characterizing the lethal dose for a specific task:

---

**Lethal Dose Conjecture** (informal)

For a given task, if the most data-efficient learner takes at least $n$ clean samples for an accurate prediction, then when a potentially poisoned, size-$N$ training set is given, any defense can tolerate at most $\Theta(N/n)$ poisoned samples while ensuring accuracy.

---

Notably, the conjecture considers an adversary who 1. knows the underlying distribution of clean data and 2. can both insert and remove samples when conducting data poisoning. The formal statement of the conjecture is in Section 4.1. We prove the conjecture in multiple settings including Gaussian classifications and offer a general perspective through distribution discrimination.

Theoretically, the conjecture relates learning from poisoned data with learning from clean data, reducing robustness against data poisoning to data efficiency and offering us a much more intuitive way to estimate the upper bound for robustness: To find out how many poisoned samples are tolerable, one can now instead search for data-efficient learners.

In addition, the conjecture implies that Deep Partition Aggregation (DPA) [20] and its extension, Finite Aggregation (FA) [34] are (asymptotically) optimal: DPA and FA are recent approaches for provable defenses against data poisoning, predicting through the majority vote of many base models trained from different subsets of training set using a given base learner. If we have the most data-efficient learner, DPA and FA can turn it into defenses against data poisoning that approach the upper bound of robustness indicated by Lethal Dose Conjecture within a constant factor.

The optimality of DPA and FA (indicated by the conjecture) outlines a practical approach to developing stronger defenses by finding more data-efficient learners. As a proof of concept, we show on CIFAR-10 and GTSRB that by simply using different data augmentations to improve the data efficiency of base learners, the certified robustness of DPA can be respectively **doubled** and **tripled** without sacrificing accuracy, highlighting the potential of this practical approach.

Another implication from the conjecture is that a stronger defense than DPA (which is asymptotically optimal assuming the conjecture) implies the existence of a more data-efficient learner. As an example, we show how to derive a learner from the nearest neighbors defenses [16], which are more robust than DPA *in their evaluation*. With the derived learner, DPA offers similar robustness to their defenses.

In summary, our contributions are as follows:

- We propose **Lethal Dose Conjecture**, characterizing the largest amount of poisoned samples any defense can tolerate for a **specific** task;
- We prove the conjecture in multiples cases including Isotropic Gaussian classifications;
- We offer a general perspective supporting the conjecture through distribution discrimination;
- We showcase how more data-efficient learners can be optimally (assuming the conjecture) transferred into stronger defenses: By simply using different augmentations to get more data-efficient base learners, we **double** and **triple** the certified robustness of DPA respectively on CIFAR-10 and GTSRB without sacrificing accuracy;
- We illustrate how a learner can be derived from the nearest neighbors defenses [16], which, in their evaluation, are much more robust than DPA (which is asymptotically optimal assuming the conjecture)—Given the derived learner, DPA transfers it to a defense with comparable robustness to the nearest neighbors defenses.

## 2  Related Work

DPA [20] and FA [34] we discuss in Section 7 are pointwise certified defenses against data poisoning, where the prediction on every sample is guaranteed unchanged within a certain attack size. Jia et al. [17] and Chen et al. [4] offer similar but probabilistic pointwise guarantees. Meanwhile, Weber et al.

[35] and Rosenfeld et al. [26] use randomized smoothing to provably defend against backdoor attacks [35] and label-flipping attacks [26], Diakonikolas et al. [10] provably approximates clean models assuming certain clean data distributions, Awasthi et al. [2] and Balcan et al. [3] provably learn linear separators over isotropic log-concave distributions under data poisoning.

Prior to Lethal Dose Conjecture, Gao et al. [11] use the framework of PAC learning to explore how the number of poisoned samples and the size of the training set can affect the threat of data poisoning attacks. Let $N$ be the size of the training set and $m$ be the number of poisoned samples. Their main results suggest that when the number of poisoned samples $m$ scales sublinearly with the size of training set $N$ (i.e. $m = o(N)$ or $m/N \to 0$), the poisoning attack will be defendable. Meanwhile, our Lethal Dose Conjecture offers a more accurate characterization: The threshold for when poisoning attacks can be too strong to be defended is when $m/N \approx \Omega(1/n)$, where $n$ is the number of samples needed by the most data-efficient learners to achieve accurate predictions.

# 3  Background and Notation

**Classification Problem**: A classification problem $(X, Y, \Omega, P, \mathcal{F})$ consists of: $X$—the space of inputs; $Y$—the space of labels; $\Omega$—the space of all labeled samples $X \times Y$; $P$—the distribution over $\Omega$ that is unknown to learners; and $\mathcal{F}$—the set of plausible learners.

**Learner**: For a classification problem $(X, Y, \Omega, P, \mathcal{F})$, a learner $f \in \mathcal{F}$ is a (stochastic) function $f : \Omega^{\mathbb{N}} \to C$ mapping a (finite) training set to a classifier, where $C$ denotes the set of all classifiers. A classifier is a (stochastic) function from the input space $X$ to the label set $Y$, where the outputs are called predictions. For a learner $f$, $f_D$ denotes the classifier corresponding to a training set $D \in \Omega^{\mathbb{N}}$ and $f_D(x)$ denotes the prediction of the classifier for input $x \in X$.

**Clean Learning**: In clean learning, given a classification problem $(X, Y, \Omega, P, \mathcal{F})$, a learner $f \in \mathcal{F}$ and a training set size $n$, the learner has access to a clean training set containing $n$ i.i.d. samples from $P$ and thus the resulting classifier can be denoted as $f_{D_n}$ where $D_n \sim P^n$ is the size-$n$ training set.

**Poisoned Learning**: In poisoned learning, given a classification problem $(X, Y, \Omega, P, \mathcal{F})$, a learner $f \in \mathcal{F}$, a training set size $N$ and a transform $T : \Omega^{\mathbb{N}} \to \Omega^{\mathbb{N}}$, the learner has access to a poisoned training set obtained by applying $T$ to the clean training set and thus the resulting classifier can be denoted as $f_{T(D_N)}$ where $D_N \sim P^N$. The symmetric distance between the clean training set and the poisoned training set, i.e. $|T(D_N) - D_N| = |(T(D_N) \setminus D_N) \cup (D_N \setminus T(D_N))|$, is called the *attack size*, which corresponds to the minimum total number of insertions and removals needed to change one training set to the other.

**Total Variation Distance**: The total variation distance between two distributions $U$ and $V$ over the sample space $\Omega$ is $\delta(U, V) = \max_{A \subseteq \Omega} |U(A) - V(A)|$, which denotes the largest difference of probabilities on the same event for $U$ and $V$.

# 4  Lethal Dose Conjecture

## 4.1  The Conjecture

Below we present a more formal statement of Lethal Dose Conjecture:

---

**Lethal Dose Conjecture**

For a classification problem $(X, Y, \Omega, P, \mathcal{F})$, let $x_0 \in X$ be an input and $y_0 = \arg\max_y P(y|x_0)$ be the maximum likelihood prediction.

- Let $n$ be the smallest training set size such that there exists a learner $f \in \mathcal{F}$ with $Pr[f_{D_n}(x_0) = y_0] \geq \tau$ for some constant $\tau$;

- For any given training set size $N$, and any learner $f \in \mathcal{F}$, there is a mapping $T : \Omega^{\mathbb{N}} \to \Omega^{\mathbb{N}}$ with $Pr[f_{T(D_N)}(x_0) = y_0] \leq 1/|Y|$ while $\mathbb{E}[|T(D_N) - D_N|] \leq \Theta(1/n) \cdot N$.

---

Informally, if the most data-efficient learner takes at least $n$ clean samples for an accurate prediction, then when a potentially poisoned, size-N training set is given, any defense can tolerate at most $\Theta(N/n)$ poisoned samples while ensuring accuracy.

**Intuitively**, if one needs $n$ samples to have a basic understanding of the task, then when $\Theta(N/n)$ samples are poisoned in a size-$N$ training set, sampling $n$ training samples will in expectation contain some poisoned samples, preventing one from learning the true distribution. **Theoretically**, the conjecture offers an intuitive way to think about or estimate the upper bound for robustness for a given task: To find out how many poisoned samples are tolerable, one now considers data efficiency instead. Practical implications are included in Section 7.

Notably, this conjecture characterizes the vulnerability to poisoning attacks of every single test sample $x_0$ rather than a distribution of test samples. While the latter type (i.e. distributional argument) is more common, a pointwise formulation is in fact more desirable and more powerful.

Firstly, a pointwise argument can be easily converted into a distributional one, but the reverse is difficult. Given a distribution of $x_0$ and the (pointwise) 'lethal dose' for each $x_0$, one can define the distribution of the 'lethal dose' and its statistics as the distributional 'lethal dose'. However, it is hard to uncover the 'lethal dose' for each $x_0$ from distributional arguments.

Secondly, samples are not equally difficult in most if not all applications of machine learning: To achieve the same level of accuracy on different test samples, the number of training samples required can also be very different. For example, on MNIST, which is a task to recognize handwritten digits, samples of digits '1' are usually easier for models to learn and predict accurately, while those of digits '6', '8' and '9 are harder as they can look more alike. In consequence, we do not expect them to be equally vulnerable to data poisoning attacks. Compared to a distributional one, the pointwise argument better incorporates such observations.

## 4.2 Proving The Conjecture in Examples

In this section we present two examples of classification problems, where we can precisely prove Lethal Dose Conjecture. For coherence, we defer the proofs to Appendix A, B, C and D.

### 4.2.1 Bijection Uncovering

**Definition 1** (Bijection Uncovering). *For any $k$, Bijection Uncovering is a k-way classification task $(X, Y, \Omega, P, \mathcal{F})$ defined as follows:*

- *The input space $X$ and the label set $Y$ are both finite sets of size $k$ (i.e. $|X| = |Y| = k$);*

- *The true distribution $P$ corresponds to a bijection $g$ from $X$ to $Y$ (note that $g$ is unknown to learners), and $\forall x \in X, \forall y \in Y, \ P(x, y) = \mathbb{1}[g(x) = y]/k$;*

- *For $y, y' \in Y$, let $T_{y \leftrightarrow y'} : \Omega^{\mathbb{N}} \to \Omega^{\mathbb{N}}$ be a transform that exchanges all labels $y$ and $y'$ in a the given training set (i.e. if a sample is originally labeled $y$, its new label will become $y'$ and vice versa). The set of plausible learners $\mathcal{F}$ contains all learners $f$ such that $Pr[f_D(x_0) = y] = Pr[f_{T_{y \leftrightarrow y'}(D)}(x_0) = y']$ for all $y, y' \in Y$ and $D \in \Omega^{\mathbb{N}}$.*

This is the setting where there is a one-to-one correspondence between inputs and classes. This is considered the 'easiest' classification as it describes the case of solving a k-way classification given a pre-trained, perfect feature extractor that puts samples from the same class close to each others and samples from different classes away from each others. In this case, since one knows whether two samples belong to the same class or not, samples can be divided into $k$ clusters and the task is essentially uncovering the bijection between clusters and class labels.

**Intuitions for The Set of Plausible Learners** $\mathcal{F}$**.** The set of plausible learners $\mathcal{F}$ is a task-dependent set and we introduce it to make sure that the learner indeed depends on and learns from training data. Here we explain in detail the set $\mathcal{F}$ in definition 1: The set of plausible learners $\mathcal{F}$ contains all learners $f$ such that $Pr[f_D(x_0) = y] = Pr[f_{T_{y \leftrightarrow y'}(D)}(x_0) = y']$ for all $y, y' \in Y$ and $D \in \Omega^{\mathbb{N}}$. Intuitively, it says that if one rearranges the labels in the training set, the output distribution will change accordingly. For example, say originally we define class 0 to be cat, and class 1 to be dog, and all dogs in the training set are labeled 0 and cats are labeled 1. In this case, for some cat image $x_0$, a learner $f$ predicts 0 with a probability of 70% and predicts 1 with a probability of 30%. What

happens if we instead define class 1 to be cat, and class 0 to be dog? Dogs in the training set will be labeled 1 and cats will be labeled 0 (i.e. label 0 and label 1 in the training set will be swapped). If $f$ is a plausible learner, meaning that it learns the association between inputs and outputs from the dataset, we expect the output distribution to change accordingly, i.e. now $f$ will predict 1 with a probability of 70% and predict 0 with a probability of 30%. Consequently, an example of a learner that is not plausible is as follow: A learner that always predicts 0 regardless of the training set, regardless of whether we associate 0 with dog or with cat.

**Lemma 1** (Clean Learning of Bijection Uncovering). *In Bijection Uncovering, given a constant $\tau \in (1/2, 1)$, for any input $x_0 \in X$ and the corresponding maximum likelihood prediction $y_0 = \arg\max_y P(y|x_0)$, let $n$ be the smallest training set size such that there exists a learner $f \in \mathcal{F}$ with $Pr[f_{D_n}(x_0) = y_0] \geq \tau$. Then $n \geq \Theta(k)$, i.e. the most data-efficient learner takes at least $\Theta(k)$ clean samples for an accurate prediction.*

**Lemma 2** (Poisoned Learning of Bijection Uncovering). *In Bijection Uncovering, for any input $x_0 \in X$ and the corresponding maximum likelihood prediction $y_0 = \arg\max_y P(y|x_0)$, for any given training set size $N$, and any learner $f \in \mathcal{F}$, there is a mapping $T : \Omega^{\mathbb{N}} \to \Omega^{\mathbb{N}}$ with $Pr[f_{T(D_N)}(x_0) = y_0] \leq 1/|Y|$ while $\mathbb{E}[|T(D_N) - D_N|] \leq \Theta(1/k) \cdot N$, i.e. poisoning $\Theta(1/k)$ of the entire training set is sufficient to break any defense.*

From Lemma 1 and Lemma 2, we see for Bijection Uncovering, the most data-efficient learner takes at least $\Theta(k)$ clean samples to predict accurately and any defense can tolerate at most $\Theta(1/k)$ of the training set being poisoned. The two quantities are **inversely proportional**, just as indicated by Lethal Dose Conjecture.

### 4.2.2 Instance Memorization

**Definition 2** (Instance Memorization). *For any $k$ and $m$, Instance Memorization is a k-way classification task $(X, Y, \Omega, P, \mathcal{F})$ defined as follows:*

- *The input space $X$ and the label set $Y$ are both finite with $|X| = m$ and $|Y| = k$;*

- *The true distribution $P$ corresponds to a mapping $g$ from $X$ to $Y$ (note that $g$ is unknown to learners), and $\forall x \in X, \forall y \in Y, P(x, y) = \mathbb{1}[g(x) = y]/m$;*

- *For $y, y' \in Y$, let $T_{y \leftrightarrow y'|x_0} : \Omega^{\mathbb{N}} \to \Omega^{\mathbb{N}}$ be a transform that exchanges labels $y$ and $y'$ for all samples with an input $x_0$ in a given training set (i.e. all $(x_0, y)$ will become $(x_0, y')$ and vice versa). The set of plausible learners $\mathcal{F}$ contains all learners $f$ such that $Pr[f_D(x_0) = y] = Pr[f_{T_{y \leftrightarrow y'|x_0}(D)}(x_0) = y']$ for all $y, y' \in Y$ and $D \in \Omega^{\mathbb{N}}$.*

This is the setting where labels for different inputs are independent and learners can only predict through memorization. This is considered the 'hardest' classification problem in a sense that the inputs are completely uncorrelated and no information is shared by labels of different inputs.

**Lemma 3** (Clean Learning of Instance Memorization). *In Instance Memorization, given a constant $\tau \in (1/k, 1)$, for any input $x_0 \in X$ and the corresponding maximum likelihood prediction $y_0 = \arg\max_y P(y|x_0)$, let $n$ be the smallest training set size such that there exists a learner $f \in \mathcal{F}$ with $Pr[f_{D_n}(x_0) = y_0] \geq \tau$. Then $n \geq \Theta(m)$, i.e. the most data-efficient learner takes at least $\Theta(m)$ clean samples for an accurate prediction.*

**Lemma 4** (Poisoned Learning of Instance Memorization). *In Instance Memorization, for any input $x_0 \in X$ and the corresponding maximum likelihood prediction $y_0 = \arg\max_y P(y|x_0)$, for any given training set size $N$, and any learner $f \in \mathcal{F}$, there is a mapping $T : \Omega^{\mathbb{N}} \to \Omega^{\mathbb{N}}$ with $Pr[f_{T(D_N)}(x_0) = y_0] \leq 1/|Y|$ while $\mathbb{E}[|T(D_N) - D_N|] \leq \Theta(1/m) \cdot N$, i.e. poisoning $\Theta(1/m)$ of the entire training set is sufficient to break any defense.*

From Lemma 3 and Lemma 4, we observe that for Instance Memorization, the most data-efficient learner takes at least $\Theta(m)$ clean samples to predict accurately and any defense can tolerate at most $\Theta(1/m)$ of the training set being poisoned. The two quantities are, **again, inversely proportional**, which is consistent with Lethal Dose Conjecture. This is likely no coincidence and more supports are presented in the following Sections 5 and 6.

# 5 An Alternative View from Distribution Discrimination

In this section, we offer an alternative view through the scope of Distribution Discrimination. Intuitively, if for two plausible distribution $U$ and $V$ over $\Omega$, the corresponding optimal predictions on some $x_0$ are different (i.e. $\arg\max_y Pr[U(y|x_0)] \neq \arg\max_y Pr[V(y|x_0)]$), then a learner must (implicitly) discriminate these two distributions in order to predict correctly. With this in mind, we present the following theorems. The proofs are respectively included in Appendix E and F.

**Theorem 1.** *Given two distributions $U$ and $V$ over $\Omega$, for any function $f : \Omega^{\mathbb{N}} \to \{0, 1\}$, we have*

$$\mathbb{E}_{D\sim U^n}[f(D)] - \mathbb{E}_{D\sim V^n}[f(D)] \leq n \cdot \delta(U, V),$$

*where $\delta(U, V)$ is the total variation distance between $U$ and $V$. Thus for a constant $\tau$, $\mathbb{E}_{D\sim U^n}[f(D)] - \mathbb{E}_{D\sim V^n}[f(D)] \geq \tau$ implies $n \geq \tau/\delta(U, V) = \Theta(1/\delta(U, V))$.*

**Theorem 2.** *Given two distributions $U$ and $V$ over $\Omega$, for any function $f : \Omega^{\mathbb{N}} \to \{0, 1\}$, there is a stochastic transform $T : \Omega^{\mathbb{N}} \to \Omega^{\mathbb{N}}$, such that $T(U^N)$ is the same distribution as $V^N$ (thus $\mathbb{E}_{D\sim U^N}[f(T(D))] - \mathbb{E}_{D\sim V^N}[f(D)] = 0$) and $\mathbb{E}_{D\sim U^N}[||T(D) - D||] = \delta(U, V) \cdot N$.*

Theorem 1 implies that to discriminate two distributions with certain confidence, the number of clean samples required is at least $\Theta(1/\delta(U, V))$, which is proportional to the inverse of their total variation distance; Theorem 2 suggests that for a size-$N$ training set, the adversary can make the two distribution indistinguishable given the ability to poison $\delta(U, V) \cdot N$ samples in expectation. This is consistent with the scaling rule suggested by the Lethal Dose Conjecture. We will also see in Section 6 how the theorems help the analysis of classification problems.

# 6 Verifying Lethal Dose Conjecture in Isotropic Gaussian Classification

In this section, we analyze the case where data from each class follow a multivariate Gaussian distribution and we will prove the very same scaling rule stated by Lethal Dose Conjecture.

**Definition 3** (Isotropic Gaussian Classification). *For any $k$ and $d$, Isotropic Gaussian Classification is a $k$-way classification task $(X, Y, \Omega, P, \mathcal{F})$ defined as follows:*

- *The input space $X$ is $\mathbb{R}^d$ and the label set $Y$ is a finite set of size $k$ (i.e. $|Y| = k$);*

- *For $y \in Y$, $\mu_y \in \mathbb{R}^d$ denotes the (unknown) center of class $y$ and the true distribution $P$ is:*

$$\begin{cases} (\forall y \in Y)\, P(y) = \frac{1}{k} & \text{Labels are uniformly distributed} \\ (\forall x \in X)(\forall y \in Y)\, P(x|y) = \frac{1}{(2\pi)^{d/2}} e^{-||x-\mu_y||^2/2} & \text{Class } y \text{ is a Gaussian } \mathcal{N}(\mu_y, I) \end{cases} \quad (1)$$

- *The set of plausible learners $\mathcal{F}$ contains all unbiased, non-trivial learners, meaning that for any $f \in \mathcal{F}$, given any $x_0$, and any plausible $P$ (i.e. the same form as Equation 1 but with potentially different $\mu_y$) where $y_0 = \arg\max_y P(y|x_0)$ is unique, we have $Pr[f_{D_n}(x_0) = y_0] > \frac{1}{k}$ and $(\forall y \neq y_0)\, Pr[f_{D_n}(x_0) = y] \leq \frac{1}{k}$ for all $n \geq 1$.*

First we present a claim regarding the optimal prediction $y_0$ corresponding to a given input $x_0$:

**Claim 1.** *In Isotropic Gaussian Classification, for any $x_0$, the corresponding maximum likelihood prediction $y_0 = \arg\max_y P(y|x_0)$ is $\arg\min_y ||x_0 - \mu_y||$.*

This follows directly from Equation 1: $y_0 = \arg\max_y P(y|x_0) = \arg\max_y \frac{1}{\sqrt{2\pi}} e^{-||x-\mu_y||^2/2} = \arg\min_y ||x - \mu_y||$. Without loss of generality and with a slight abuse of notation, we assume in the rest of Section 6 that $\mu_1$, $\mu_2$ are respectively the closest and the second closest class centers to $x_0$ and therefore $y_0 = 1$ is the optimal prediction. Let $d_1 = ||x_0 - \mu_1||$ and $d_2 = ||x_0 - \mu_2||$ be the distances of $\mu_0$ and $\mu_1$ to $x_0$. An illustration is included in Figure 1(a).

First we define a parameter $\Delta$ that will later help us to analyze both clean learning and poisoned learning of Isotropic Gaussian Classification: $\Delta$ is the total variation distance between two isotropic Gaussian distribution with centers separated by a distance of $d_2 - d_1$, i.e. $\Delta = \delta(U, V)$ for some $U = \mathcal{N}(\mu, I)$ and $V = \mathcal{N}(\mu', I)$ where $||\mu - \mu'|| = d_2 - d_1$. We introduce a lemma proved by Devroye et al. [9].

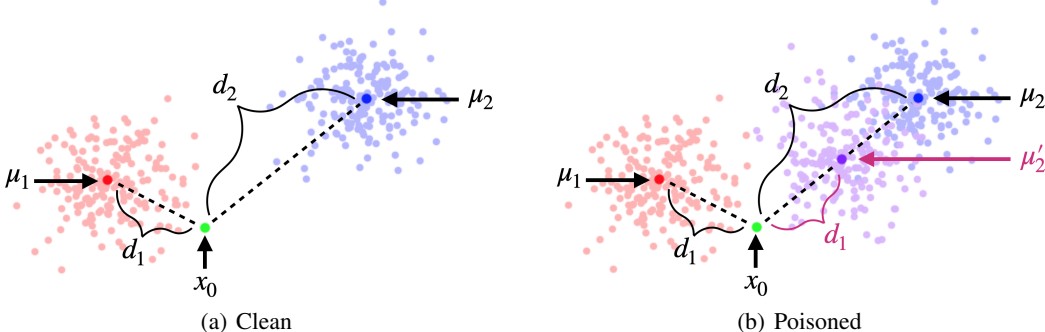

(a) Clean        (b) Poisoned

Figure 1: Illustrations of Isotropic Gaussian Classification under clean learning and poisoned learning, where $\mu_1, \mu_2$ are centers for clean training samples with labels 1 (red) and 2(blue); $\mu_2'$ is the center for the poisoned training samples with labels 2 (purple). Note that the illustrations contain only the **two closest classes** to $x_0$ and the setting contains $k \geq 2$ classes.

**Lemma 5** (Total variation distance for Gaussians with the same covariance[9]). *For two Gaussian distribution $U = \mathcal{N}(\mu, \Sigma)$ and $V = \mathcal{N}(\mu', \Sigma)$, their total variation distance is $\delta(U, V) = Pr[\mathcal{N}(0, 1) \in [-r, r]]$ where $r = \sqrt{(\mu - \mu')^T \Sigma^{-1}(\mu - \mu')}/2$.*

Thus $\Delta = Pr\left[|\mathcal{N}(0, 1)| \leq d_2 - d_1\right] = \text{erf}\left((d_2 - d_1)/\sqrt{2}\right)$ where $\text{erf}(x) = (2/\sqrt{\pi}) \cdot \int_0^x e^{-t^2} dt$ is the Gaussian error function.

**Lemma 6** (Clean Learning of Isotropic Gaussian Classification). *In Isotropic Gaussian Classification, given a constant $\tau \in (1/2, 1)$, for any input $x_0 \in X$ and the corresponding maximum likelihood prediction $y_0 = \arg\max_y P(y|x_0)$, let $n$ be the smallest training set size such that there exists a learner $f \in \mathcal{F}$ with $Pr[f_{D_n}(x_0) = y_0] \geq \tau$. Then $n \geq \Theta(k/\Delta)$, i.e. the most data-efficient learner takes at least $\Theta(k/\Delta)$ clean samples for an accurate prediction.*

**Lemma 7** (Poisoned Learning of Isotropic Gaussian Classification). *In Isotropic Gaussian Classification, for any input $x_0 \in X$ and the corresponding maximum likelihood prediction $y_0 = \arg\max_y P(y|x_0)$, for any given training set size $N$, and any learner $f \in \mathcal{F}$, there is a mapping $T : \Omega^{\mathbb{N}} \to \Omega^{\mathbb{N}}$ with $Pr[f_{T(D_N)}(x_0) = y_0] \leq 1/|Y|$ while $\mathbb{E}[|T(D_N) - D_N|] \leq \Theta(\Delta/k) \cdot N$, i.e. poisoning $\Theta(\Delta/k)$ of the entire training set is sufficient to break any defense.*

The proofs of Lemma 6 and Lemma 7 are included respectively in Appendix G and H. Intuitively, what we do is to construct a second, perfectly legit distribution that is not far from the original one (measured with the total variation distance), so that any classifier must either fail on the original one or fail on the one we construct. If it fails on the original one, the adversary achieves its goal even without poisoning the training set. If it fails on the one we construct, the adversary can still succeed by poisoning only a limited fraction of the training set because the distribution we construct is close to the original one (measured with total variation distance).

Through the lemmas, we show that for Isotropic Gaussian Classification, the most data-efficient learner takes at least $\Theta(k/\Delta)$ clean samples to predict accurately and any defense can be broken by poisoning $\Theta(\Delta/k)$ of the training set, once again matching the statement of Lethal Dose Conjecture.

## 7 Practical Implications

### 7.1 (Asymptotic) Optimality of DPA and Finite Aggregation

In this section, we highlight an important implication of the conjecture: Deep Partition Aggregation [20] and its extension, Finite Aggregation [34] are (asymptotically) optimal—Using the most data-efficient learner, they construct defenses approaching the upper bound of robustness indicated by Lethal Dose Conjecture within a constant factor.

**Deep Partition Aggregation [20]:** DPA predicts through the majority votes of base learners trained from disjoint data. Given the number of partitions $k$ as a hyperparameter, DPA is a learner constructed

using a deterministic base learner $f$ and a hash function $h : X \times Y \to [k]$ mapping labeled samples to integers between 0 and $k - 1$. The construction is:

$$\text{DPA}_D(x_0) = \arg\max_y \text{DPA}_D(x_0, y) = \arg\max_{y \in \mathcal{Y}} \frac{1}{k} \sum_{i=0}^{k-1} \mathbb{1}\left[f_{P_i}(x_0) = y\right],$$

where $P_i = \{(x, y) \in D \mid h(x, y) = i\}$ is a partition containing all training samples with a hash value of $i$ and $\text{DPA}_D(x_0, y) = \frac{1}{k} \sum_{i=0}^{k-1} \mathbb{1}\left[f_{P_i}(x_0) = y\right]$ denotes the average votes count for class $y$. Ties are broken by returning the smaller class index in $\arg\max$.

**Theorem 3** (Certified Robustness of DPA against Data Poisoning [20]). *Given a training set $D$ and an input $x_0$, let $y_0 = DPA_D(x_0)$, then for any training set $D'$ with*

$$|D - D'| \leq \frac{k}{2}\left(DPA_D(x_0, y_0) - \max_{y \neq y_0}\left(DPA_D(x_0, y) + \frac{\mathbb{1}\left[y < y_0\right]}{k}\right)\right) \tag{2}$$

*we have $DPA_D(x_0) = DPA_{D'}(x_0)$, meaning the prediction remains unchanged with data poisoning.*

Let $n$ be the average size of partitions, i.e. $n = \sum_{i=0}^{k-1} |P_i|/k = N/k$ where $N = |D|$ is the size of the training set. Assuming the hash function $h : X \times Y \to [k]$ uniformly distributes samples into different partitions, we have $\text{DPA}_D(x_0, y) \approx Pr[f_{D_n}(x_0) = y]$ and therefore the right hand side of Equation 2 approximates

$$\frac{Pr[f_{D_n}(x_0) = y_0] - \max_{y \neq y_0} Pr[f_{D_n}(x_0) = y]}{2n} \cdot N = \Theta\left(\frac{1}{n}\right) \cdot N$$

when the base learner $f$ offers a margin $\Delta = Pr[f_{D_n}(x_0) = y_0] - \max_{y \neq y_0} Pr[f_{D_n}(x_0) = y] \geq \tau$ with $n$ samples for some constant $\tau$. When $f$ is the most data-efficient learner taking $n$ clean samples with a margin $\Delta \geq \tau$ for an input $x_0$, then given a potentially poisoned, size-$N$ training set is given, DPA (using $f$ as the base learner) tolerates $\Theta\left(N/n\right)$ poisoned samples, approaching the upper bound of robustness indicated by Lethal Dose Conjecture within a constant factor.

The analysis for Finite Aggregation (FA) is exactly the same by substituting Theorem 3 with the certified robustness of FA. For details, please refer to Theorem 2 by Wang et al. [34].

## 7.2 Better Learners to Stronger Defenses

Assuming the conjecture is true, we have in our hand the (asymptotically) optimal approaches to convert data-efficient learners to strong defenses against poisoning. This reduces developing stronger defenses to finding more data-efficient learners. In this section, as a proof of concept, we show that simply using different data augmentations can increase the data efficiency of base learners and **vastly** improve (*double* or even *triple*) the certified robustness of DPA, essentially highlighting the potential of finding data-efficient learners in defending against data poisoning, which has been a blind spot for our community.

**Setup.** Following Levine and Feizi [20], we use Network-In-Network[21] architecture for all base learners and evaluate on CIFAR-10[18] and GTSRB[29]. For the baseline (DPA_baseline), we follow exactly their augmentations, learning rates (initially 0.1, decayed by a factor of $1/5$ at $30\%$, $60\%$, and $80\%$ of the training process), batch size (128) and total epochs (200). For our results (DPA_aug0 and DPA_aug1 on CIFAR-10; DPA_aug on GTSRB), we use predefined AutoAugment [6] policies for data augmentations, where DPA_aug0 uses the policy for CIFAR-10, DPA_aug1 uses the policy for Imagenet and DPA_aug uses the policy for SVHN, all included in torchvision[24]. We use an initial learning rate of 0.005, a batch size of 16 for 600 epochs on CIFAR-10 and 1000 epochs on GTSRB.

**Evaluation.** First, we evaluate the test accuracy of base learners with limited data (i.e. using $1/k$ of the entire training set of CIFAR-10 and GTSRB where $k$ ranges from 50 to 500). In Figure 2(a), simply using AutoAugment greatly improves the accuracy of base learners with limited data: On CIFAR-10, with both augmentation policies, the augmented learners achieve similar accuracy as the baseline using only $1/2$ of data (i.e. with $k$ that is twice as large); On GTSRB, the augmented learner achieves similar accuracy as the baseline using only $1/4 \sim 1/3$ of data.

Now we use more data-efficient learners in DPA to construct stronger defenses. For baselines, we use DPA with $k = 50, 100, 250$ on CIFAR-10 and $k = 50, 100$ on GTSRB, defining essentially 5

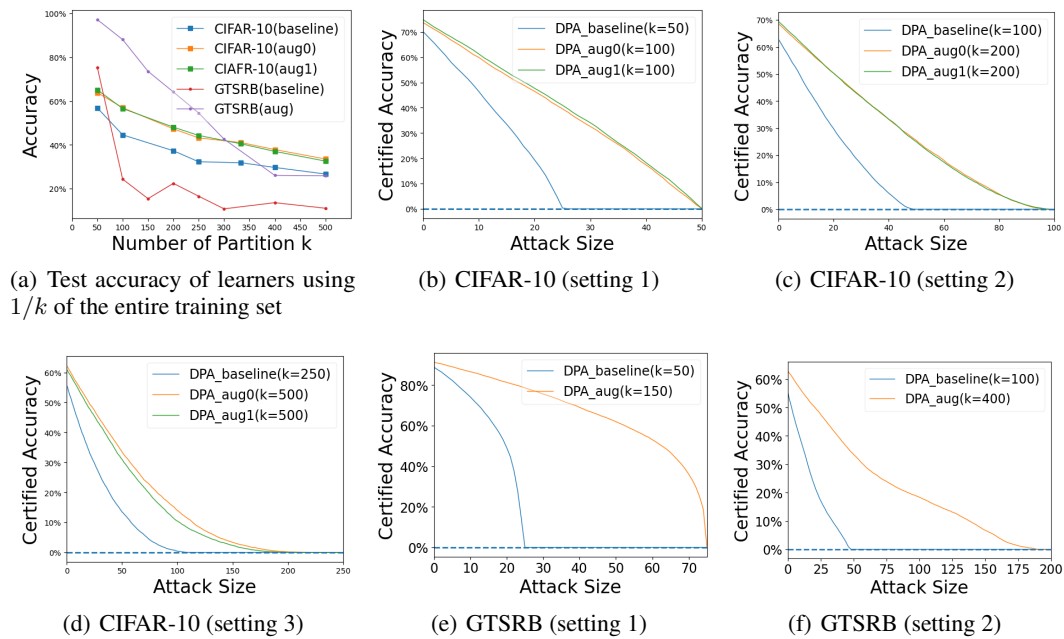

(a) Test accuracy of learners using $1/k$ of the entire training set

(b) CIFAR-10 (setting 1)

(c) CIFAR-10 (setting 2)

(d) CIFAR-10 (setting 3)

(e) GTSRB (setting 1)

(f) GTSRB (setting 2)

Figure 2: Experiments that construct stronger defenses against data poisoning by using more data-efficient base learners for DPA. By simply using different data augmentations to improve the data efficiency of base learners, the certified robustness can be respectively **doubled** and **tripled** on CIFAR-10 and GTSRB without sacrificing accuracy.

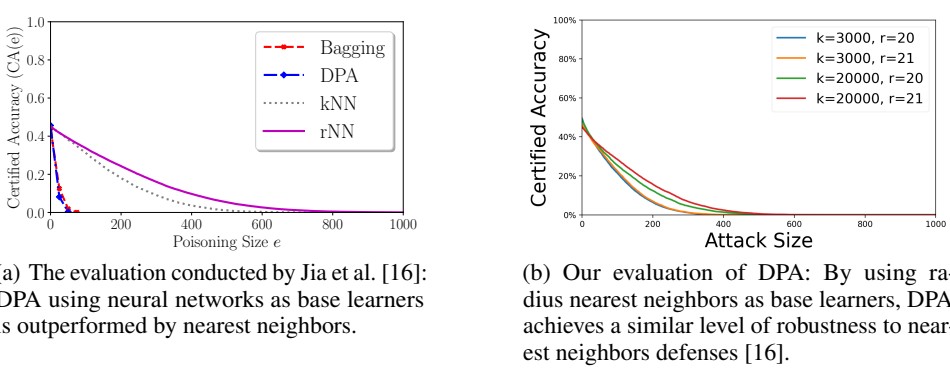

(a) The evaluation conducted by Jia et al. [16]: DPA using neural networks as base learners is outperformed by nearest neighbors.

(b) Our evaluation of DPA: By using radius nearest neighbors as base learners, DPA achieves a similar level of robustness to nearest neighbors defenses [16].

Figure 3: Comparing the certified robustness of DPA with the nearest neighbor defenses[16] on CIFAR-10, suggesting kNN and rNN are no stronger defense than DPA but are using a more data-efficient learner given the accuracy requirements.

settings. When using augmented learners in DPA, we increase $k$ accordingly so that the accuracy of base learners remains at the same level as the baselines, as indicated by Figure 2(a). On CIFAR-10, in Figure 2(b), 2(c) and 2(d), the attack size tolerated by DPA is roughly doubled using both augmented learners for similar certified accuracy. On GTSRB, in Figure 2(e) and 2(f), the attack size tolerated by DPA is more than tripled using the augmented learner for similar certified accuracy.

The improvements are quite significant, corroborating the potential of this practical approach—developing stronger defenses against poisoning through finding more data-efficient learners.

### 7.3 Better Learners from Stronger Defenses

Assuming the conjecture is true, when some defense against data poisoning is clearly more robust than DPA (even in restricted cases), there should be a more data-efficient learner. Jia et al. [16]

propose to use nearest neighbors, i.e. kNN and rNN, to certifiably defend against data poisoning. Interestingly, in their evaluations, kNN and rNN tolerate much more poisoned samples than DPA when the required clean accuracy is low, as shown in Figure 3(a). Here we showcase how a learner is derived from their defense, which, when being used as base learners, boosts the certified robustness of DPA to a similar level to the nearest neighbors defenses.

In [16], histogram of oriented gradients (HOG) [7] is used as the predefined feature space to estimate the distance between samples. Consequently, we use *radius nearest neighbors* over HOG features as the base learner of DPA. Given a threshold $r$, radius nearest neighbors find the majority votes of training samples within an $\ell_1$ distance of $r$ to the test sample. When there is no training sample with a distance of $r$, we let the base learner output a token $\perp$ denoting outliers so that it will not affect the aggregation. The results are included in Figure 3(b), where DPA offers similar robustness curves as kNN and rNN in Figure 3(a) by using the aforementioned base learner and hence the same prior.

## 8 Conclusion

In this work, we propose **Lethal Dose Conjecture**, which characterizes the largest amount of poisoned samples any defense can tolerate for a specific task. We prove the conjecture for multiple cases and offer general theoretical insights through distribution discrimination. The conjecture implies the (asymptotic) optimality of DPA [20] and FA [34] in a sense that they can transfer the most data-efficient learners to one of the most robust defenses, revealing a practical approach to obtaining stronger defenses via improving data efficiency of (non-robust) learners. Empirically, as a proof of concepts, we show that simply using different data augmentations can increase the data efficiency of base learners, and therefore respectively **double** and **triple** the certified robustness of DPA on CIFAR-10 and GTSRB. This highlights both the importance of Lethal Dose Conjecture and the potential of the practical approach in searching for stronger defenses.

## Acknowledgements

This project was supported in part by NSF CAREER AWARD 1942230, a grant from NIST 60NANB20D134, HR001119S0026 (GARD), ONR YIP award N00014-22-1-2271, Army Grant No. W911NF2120076 and the NSF award CCF2212458.

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
