# OpenReview forum: "Lethal Dose Conjecture on Data Poisoning"
_NeurIPS.cc/2022/Conference — NeurIPS 2022 Accept_

### Official Review · Reviewer_a652 · 2022-07-05

**Rating:** 6
**Confidence:** 4
**Soundness:** 4 excellent
**Presentation:** 2 fair
**Contribution:** 3 good

**Summary:**

The paper suggests a conjecture that quantifies what is the maximal fraction of training data that might be poisoned, while achieving a given required accuracy rate $\epsilon$. Specifically, the conjecture is that roughly $1/n$ of the training data is the maximal poisoned data fraction, where $n$ is the sample complexity of the task, that is, the sample size required to achieve accuracy $\epsilon$ in a "clean" environment (without presence of an adversary). The conjecture is proved for some special cases such as instance memorization. Another theoretical aspect of the paper is a general view of the poisoning problem through distribution discrimination. Practically, the paper summarizes results of some experiments supporting the conjecture: On one hand, robust learners can be derived from data-efficient learners, and on the other hand, data-efficient learners can be derived from robust learners.

**Questions:**

1. In the definition of a learner, and poisoned learning: The domain of $T$ and $f$ seems to be defined as infinite vectors where I guess it should be finite vectors?
2. The formulation of the formal statement of the conjecture in page 3 is not justified enough, in my opinion. It seems that the conjecture is formulated with respect to a specific given data point $x_0$. I guess that this what a "specific task" (as written in the introduction) means? However, a "specific task" might be understood as drawing the test point from a *specific* hidden marginal distribution over instances, as usually done in PAC learning. Also, isn't this suggested formulation might be better? For example, think of a point that can only suffer attacks of a very small size, but on the other hand is not likely to be drawn as a test point. Isn't it better to define the lethal dose to be higher, than what reflects in the conjecture, in this case? (because a wrong prediction on this point is not lethal).
3. In definition 1: What does "plausible learners" mean? In what sense are they plausible?
4. The paragraph that comes after definition 1 is not clear to me. What are the "classes" here? In what sense is this setting the easiest?
5. Isn't the proof of Lemma 1 is just by the fact that the VC-dimension of the corresponding hypothesis class is $k$? Just think of the $k$ labels as all possible binary labelings of $\log_2(k)$ data points.
6. The paragraph that comes after definition 2 is not clear to me. Why is this setting so much harder compared with the setting of definition 1? Also, definition 2 seems like a generalization of definition 1, and if that is indeed the case,  perhaps it is good to mention that.

**Limitations:**

Yes, except for what is written in question 2 about the formulation of the conjecture.

**Strengths And Weaknesses:**

Strengths:

1. The conjecture sounds reasonable. The informal arguments as well as the experiments supporting it are quite convincing.
2. The "distribution discrimination" view of the problem is intuitive and insightful.
3. The practical technique of using data augmentation to increase accuracy is interesting and insightful.
4. The idea of deriving data efficient-learners from robust learners is interesting as well.

Weaknesses:
1. The absence of some standard notions in machine learning that seems related to the paper makes the paper and its contribution harder to understand.  For example, it seems that the terms "sample complexity" and "realizability" should have been integrated in the basic definitions. Another example is Lemma 1 which seemingly can be simply proved using the well known VC-dimension, which is not mentioned anywhere in the paper (elaboration in  question 5).
2. The theoretical contribution of the paper is not clear enough to me: The formal statement of the conjecture is not justified enough in my opinion (elaboration in question 2). It is also not explained why the special cases described in Section 4 are interesting, and in what sense they are "easy" or "hard", as written in the paper (see elaboration in questions 4,6).
3. There are some unclear parts in the text, which also makes it hard to evaluate the paper's contribution. Examples can be found in questions 1,3.

---

> ### Author Response · Authors · 2022-08-02
> **Response to Reviewer a652: part 1**
>
> Thank you so much for spending time reviewing our work! We value every feedback from you and will try our best to answer your questions.
>
> **(1)** Weakness: ‘The absence of some standard notions in machine learning that seems related to the paper makes the paper and its contribution harder to understand. For example, it seems that the terms "sample complexity" and "realizability" should have been integrated in the basic definitions.  ’
>
> **Answer:**
> Thanks for the suggestion! We will be considering how we can incorporate them to improve the presentation of our work.
>
>
> **(2)** Question: ‘In the definition of a learner, and poisoned learning: The domain of T
>  and f seems to be defined as infinite vectors where I guess it should be finite vectors?’
>
>
> **Answer:**
> Good question!
> Here $\mathbb{N}$ denotes the set of all natural numbers and $\Omega^\mathbb{N}$, the domain of T and f, is the set of all *finite* datasets. In other words, we consider that the training set can contain an arbitrary but finite number of samples, meaning that the size of the training set can be 10, 10^4, 10^10… **it can be arbitrarily large, but not infinite**.
>
>
> **(3)** Question: ‘The formulation of the formal statement of the conjecture in page 3 is not justified enough, in my opinion. It seems that the conjecture is formulated with respect to a specific given data point $x_0$. I guess that this what a "specific task" (as written in the introduction) means? However, a "specific task" might be understood as drawing the test point from a specific hidden marginal distribution over instances, as usually done in PAC learning. Also, isn't this suggested formulation might be better? For example, think of a point that can only suffer attacks of a very small size, but on the other hand is not likely to be drawn as a test point. Isn't it better to define the lethal dose to be higher, than what reflects in the conjecture, in this case? (because a wrong prediction on this point is not lethal).’
>
> **Answer:**
> Good comment. Indeed, a ‘task’ is more often interpreted as a distributional argument rather than the pointwise one we present. However, the pointwise formulation is in fact **more desirable and more powerful**.
>
> Firstly, a pointwise argument can be easily converted into a distributional one, but the reverse is difficult. Given a distribution of $x_0$ and the (pointwise) ‘lethal dose’ for each $x_0$, one can define the distribution of the ‘lethal dose’ and its statistics as the distributional ‘lethal dose’. However, it is hard to uncover the ‘lethal dose’ for each $x_0$ from distributional arguments.
>
> Secondly, samples are not equally difficult in most if not all applications of machine learning: To achieve the same level of accuracy on different test samples, the number of training samples required can also be very different.
> For example, on MNIST, which is a task to recognize handwritten digits, samples of digits ‘1’ are usually easier for models to learn and predict accurately, while those of digits ‘6’, ‘8’ and ‘9 are harder as they can look more alike.
> In consequence, we do not expect them to be equally vulnerable to data poisoning attacks. Compared to a distributional one, the pointwise argument better incorporates such observations.
>
>
> Due to page limits, we have added discussions about this to the revised draft in Appendix J. We will move the discussion to the main paper for the camera-ready version, which allows an additional content page.

---

> > ### Author Response · Authors · 2022-08-02
> > **Response to Reviewer a652: part 2**
> >
> > **(4)** Question: ‘In definition 1: What does "plausible learners" mean? In what sense are they plausible?’
> >
> > **Answer:**
> > The set of plausible learners $\mathcal{F}$ is a task-dependent set and we introduce it to make sure that the learner indeed depends on and learns from training data.
> >
> > Here we explain in detail the set $\mathcal{F}$ in definition 1.
> >
> > Firstly, we quote the entire sentence from our paper: ‘The set of plausible learners $\mathcal{F}$ contains all learners $f$ such that $Pr [f_{D}(x_0) = y] = Pr[f_{T_{y\leftrightarrow y'}(D)}(x_0) = y']$ for all $y,y'\in Y$ and $D \in \Omega^\mathbb{N}$.’
> >
> > ‘Plausible learners’ simply refers to learners $f$ such that $Pr [f_{D}(x_0) = y] = Pr[f_{T_{y\leftrightarrow y'}(D)}(x_0) = y']$ for all $y,y'\in Y$ and $D \in \Omega^\mathbb{N}$. Intuitively, it says that if one rearranges the labels in the training set, the output distribution will change accordingly.
> >
> > For example, say originally we define class 0 to be cat, and class 1 to be dog, and all dogs in the training set are labeled 0 and cats are labeled 1. In this case,  for some cat image $x_0$, a learner $f$ predicts 0 with a probability of 70% and predicts 1 with a probability of 30%.
> >
> > What happens if we instead define class 1 to be cat, and class 0 to be dog? Dogs in the training set will be labeled 1 and cats will be labeled 0 (i.e. label 0 and label 1 in the training set will be swapped). If $f$ is a plausible learner, meaning that it learns the association between inputs and outputs from the dataset, we expect the output distribution to change accordingly, i.e. now $f$ will predict 1 with a probability of 70% and predict 0 with a probability of 30%.
> >
> > Here is an example of a learner that is not plausible: A learner that always predicts 0 regardless of the training set, regardless of whether we associate 0 with dog or with cat.
> >
> > Due to page limits, we have added the explanations to the revised draft in Appendix K. We will move them into main body of our paper for the camera-ready version, where an additional page will be allowed.
> >
> > **(5)** Question: ‘The paragraph that comes after definition 1 is not clear to me. What are the "classes" here? In what sense is this setting the easiest?’
> >
> >
> > **Answer:**
> > Sorry for the confusion. Classes are associated with labels. Each class has a label and each label corresponds to a class.
> >
> > This setting is intuitively ‘easy’ because the input space or the feature space given has  nice properties that are helpful for classification: Samples are already perfectly clustered in the input space according to labels. Samples with the same label stay close while samples with different labels are away from each other, so that for every class/label, a **single** clean training sample from that class will allow one to identify **all** samples from that class.
> >
> > For example, imagine you want to solve a classification task and you are given a feature extractor that puts and only puts samples with the same label close to each other in the feature space.  You will be able to solve the classification using only one sample from each class. For a test sample, you simply find the closest one in the training set and that will be the correct label. This is an easy setting because the feature extractor given is powerful.
> >
> >
> >
> > **(6)** Question: ‘Isn't the proof of Lemma 1 is just by the fact that the VC-dimension of the corresponding hypothesis class is k? Just think of the k labels as all possible binary labelings of $\log_2(k)$ data points.’
> >
> >
> > **Answer:**
> > Sorry we haven’t figured out your constructions, but we are all very interested! Do you mind elaborating a little bit more on what the insight is for ‘think of the k labels as all possible binary labelings of $\log_2(k)$ data points’?
> >
> >
> > **(7)** Question: ‘The paragraph that comes after definition 2 is not clear to me. Why is this setting so much harder compared with the setting of definition 1? Also, definition 2 seems like a generalization of definition 1, and if that is indeed the case, perhaps it is good to mention that.’
> >
> > **Answer:**
> >
> > Similar to our answer for Question (5), this is a ‘difficult’ setting because the input space or the feature given is terrible in a sense that there is no correlation between labels corresponding to different inputs, so that one needs to see *all* samples in order to identify exactly *all* samples from a class.
> >
> > For an extreme example, imagine you want to do classification based on only hash values of images. This is truly a poor choice of features as similar (but not identical) hash values may correspond to completely unrelated samples, and it is for sure a hard task, all because the feature extractor (in this case it is the hashing function) is so terrible.
> >
> >
> > **Once again, thank you for your insightful comments! If we help address your concerns, please do consider raising your score for our work!**

---

> > > ### Comment · Reviewer_a652 · 2022-08-08
> > > **Response to authors**
> > >
> > > Response to answer for question #2:
> > >
> > > OK, this definition makes sense. I think that the notation $X^{\mathbb{N}}$ usually refers to countably infinite vectors with values from a set $X$, so perhaps consider changing the notation.
> > >
> > > Response to answer for question #3:
> > >
> > > Nice discussion and example! I think it will really help clarifying the conjecture if this discussion will appear close to it.
> > >
> > > Response to answer to question #6:
> > >
> > > I'm sorry, I think I had a mistake while thinking about this proof attempt. However, there might be an easier proof by using the DS dimension which was recently shown to characterize multiclass classification in the following paper:
> > >
> > > https://arxiv.org/abs/2203.01550
> > >
> > > I am really not sure about it, but you may have a look if you are interested.

---

> > > > ### Author Response · Authors · 2022-08-08
> > > > **Thank you!**
> > > >
> > > > We will check and see if we can figure out an easier proof from the paper you provided~
> > > >
> > > > Right now I only get to take a quick look at that paper but I feel I may learn a lot from it!
> > > >
> > > > **Thank you for your feedback and the reference!**
> > > >
> > > > **Meanwhile, if your original concerns are addressed or alleviated, please consider updating your rating for our work~**

---

> > > > > ### Comment · Reviewer_a652 · 2022-08-09
> > > > > **Thank you**
> > > > >
> > > > > Thank you very much for this clear, detailed and interesting discussion.
> > > > > In my opinion, the verbal description of the score 6 is most suitable for the paper.
> > > > > Overall I think that the paper contains significant scientific contributions to the field, and as such will be a good contribution to NeurIPS.

---

### Official Review · Reviewer_7KLu · 2022-07-08

**Rating:** 7
**Confidence:** 5
**Soundness:** 4 excellent
**Presentation:** 4 excellent
**Contribution:** 4 excellent

**Summary:**

This paper suggests a hypothesis for the necessary and sufficient amount of malicious samples needed asymptotically for successful data poisoning. Most notably, the amount is inversely proportional to the minimum amount of data needed to learn the concept for the chose model class. The hypothesis is proven on some learning scenarios. Empirically, it is shown that learning pipelines using more data efficient base learner can achieve higher certified robustness against poisoning.

**Questions:**

I have two main questions.

1. How should we interpret $n$ --- the minimum amount of data required by the most sample efficient learner? Given a distribution and a model class for the base learner, $n$ would become a constant, wouldn't it? For example, for a linear separable data distribution with margin $\epsilon$, $n$ would just be a constant for a given $\epsilon$. Now, notice the bound is asymptotic, a constant $n$ means the data poisoner always need to poison a constant fraction of the data set. Could you give an example that 1) the model class has a variable $k$, e.g. size of neurons in an NN, 2) $n$ scales with $k$, and 3) the amount of poisoning examples needed has lower order than constant portion given increasing $k$?

2. Correct me if I'm wrong: the hypothesis suggests that a more complex base learner may be more prone to data poisoning attack. On the other hand, a more complexity model (e.g. deep learning models) has the potential to fit both the poisoning data and the clean data separately, while a simple model (e.g. linear classifier) cannot. How do these two view reconcile with each other? This is out of the scope of the paper, and will not be the ground of my acceptance/rejection. But I'm curious about your opinion. Thanks.

**Limitations:**

The limitations have been adequately addressed.

**Strengths And Weaknesses:**

This paper proposes a wholistic idea with ample theoretical and empirical evidence. The message is clear and thought-provoking. I learned something new. Thank you.

In terms of clarity, this paper is of high quality. The main concepts are mostly clear stated, and the theoretical insights is coupled with intuitive explanation, making it easy to understand by audience with different levels of theoretical background.

For significance, although the paper is only a hypothesis, the hypothesis statement is thought provoking. The implication to real-world scenario is also impactful: data efficient base learner can achieve higher robustness against poisoning. The empirical evidence further increases the significance.

For originality, the paper stems from previous work of DPA and FA, the angle is new. Whether the hypothesis is eventually proven true or false, the idea is worth presenting to the machine learning community.

Overall, this paper is of high quality. However, I do have a few lingering problem about the strength of the statement, and hope the authors can clarify.

---

> ### Author Response · Authors · 2022-08-02
> **Response to Reviewer 7KLu**
>
> Thanks for your time reviewing our work! We truly appreciate your comments and will do our best to answer your questions.
>
> **(1)** Question: ‘How should we interpret n --- the minimum amount of data required by the most sample efficient learner? Given a distribution and a model class for the base learner, n would become a constant, wouldn't it? For example, for a linear separable data distribution with margin ϵ, n would just be a constant for a given ϵ. Now, notice the bound is asymptotic, a constant n means the data poisoner always need to poison a constant fraction of the data set. Could you give an example that 1) the model class has a variable
> k, e.g. size of neurons in an NN, 2) n scales with k, and 3) the amount of poisoning examples needed has lower order than constant portion given increasing k?’
>
> **Answer:**
> Firstly, Lethal Dose Conjecture suggests that **a certain fraction** will be the ‘Lethal Dose’. In another word, the maximum tolerable number of poisoning samples scales linearly with the size of the entire training set $N$. **But more importantly**, the conjecture offers a characterization of the fraction, i.e. the fraction will be $\Theta(1/n)$, where $n$ is the minimum number of samples required by the most data-efficient learner.
>
> **Please correct us if we do not understand the second half of the question accurately:** If one is using DPA and use neural networks (e.g. CNN, Transformers…), the training set for each base model (i.e. each partition) will typically have a smaller size and simply increasing model size can often lead to reduced performance. In this sense, when the size of models increase, the base learner can be less data-efficient (i.e. n increases when models get larger), and therefore the number of poisoned samples tolerated will decrease to a lower portion.
>
>
> **(2)** Question: ‘Correct me if I'm wrong: the hypothesis suggests that a more complex base learner may be more prone to data poisoning attack. On the other hand, a more complexity model (e.g. deep learning models) has the potential to fit both the poisoning data and the clean data separately, while a simple model (e.g. linear classifier) cannot. How do these two view reconcile with each other? This is out of the scope of the paper, and will not be the ground of my acceptance/rejection. But I'm curious about your opinion. Thanks.’
>
> **Answer:**
> Even assuming that a more complex base learner is more data-efficient, the conjecture does not imply that such a base learner is itself more resilient to data poisoning.
> An important implication of the conjecture is that DPA is nearly optimal in converting base learners to defenses against data poisoning, with **no robustness requirement** on base learners.
> We agree that in modern paradigms complex models are usually easier to overfit and may be more vulnerable to data poisoning attacks, but it is still too early to say that such correspondence is inevitable.
>
> **Finally, thanks again for your insights and questions! Could you please consider raising your score if we do help address the concerns?**

---

> > ### Comment · Reviewer_7KLu · 2022-08-08
> > **More discussion**
> >
> > Thank you for the responses. They are clear and I still lean towards accept the paper as it is.
> >
> > I just want to follow up my first question. I understand that the lethal dose is $\Theta(1/n)$ and the total number of poison examples scales with $N$. I want to understand, however, the complexity of $n$ itself in practice. First of all, $n$ does not necessarily scale with $N$. Rather, it is influenced by the underlying distribution, or the classification task itself. As I said in my question, if the true underlying distribution is linearly separable with a margin $\epsilon$, then $n$ will be determined by $\epsilon$ and the input dimension $d$. That is, $n$ is a constant w.r.t. $N$. The lethal dose poisoning fraction $\Theta(1/n)$ would simply mean the attacker needs to poison a constant fraction of the training set. It's true but not very useful because the $\Theta$ bound also allows variations up to a constant factor.
> >
> > So my question is, is there an example such that for a common type of model, 1) $n$ scales up very fast with the "difficulty" of the underlying tasks, and 2) $\Theta(1/n)$ diminishes very fast as a result? In particular, is there a "bad" underlying distribution such that the poisoner can easily achieve its goal with very few examples for common classifiers? (For example, a checker board where cells of different colors have different labels?)

---

> > > ### Author Response · Authors · 2022-08-08
> > > **Follow-up for Reviewer 7KLu**
> > >
> > > Sorry for this late reply, and thank you so much for the discussion!
> > >
> > > **(1)** For your concern regarding the usefulness of Lethal Dose Conjecture:
> > >
> > > **Answer:**
> > > Yes, since a constant variation is possible, when one has $N=10000$ samples and one assumes $n=500$ samples are needed by the most data-efficient learners (given one's prior), the conjecture will not be able to tell whether $20$ or $21$ poisoned samples can be tolerated. However, it for sure raises a flag when some unknown data source contributes hundreds of samples to one's training set. **Admittedly, the current conjecture is not perfect, but given its generality and flexibility, we do believe it is a very promising step towards mitigating the threat of data poisoning attacks!**
> > >
> > > **(2)** For your question regarding examples that a (test) sample $x_0$ is very 'difficult' such that the corresponding $n$, i.e. the number of samples required to predict accurately on it, is large:
> > >
> > > **Answer:**
> > > Indeed, $n$, the number of samples required by the most data-efficient learner, does **not** depend on $N$, the actual number of samples in a training set.
> > >
> > > It depends on the underlying distribution (as you mentioned), and importantly, **the prior information** one has regarding the distribution. Prior information refers to anything that one knows prior to seeing any data (which we characterize through the set of plausible learners $\mathcal{F}$ in definition 1, definition 2, and definition 3). For an extreme example, if one knows already the exact data distribution as the prior information, no sample is needed at all by the most data-efficient learner and any number of poisoned samples can be tolerated!
> > >
> > >
> > >
> > > Here is **an example**: It is a binary classification and the prior information is that data from each class follows an isotropic multivariate Gaussian distribution with an unknown mean. In this case, we know that the optimal decision boundary is simply a linear one (i.e. a hyperplane). In this very simple example, most of the (test) samples $x_0$ can be easy as they are far away from the optimal decision boundary. However, when a sample $x_0$ is close to the optimal hyperplane, a large number of training samples will be required to determine its corresponding maximum likelihood prediction (**This is a direct implication of our theoretical analysis in Section 6, by letting $(d_2 - d_1)\to 0$ in our Lemma 5, Lemma 6 and Lemma 7**). In another word, $n$ scales very fast when the $x_0$ is getting closer to the optimal boundary and the 'difficulty' is increasing.
> > >
> > > For **real-life examples**, one may consider the many tasks in natural language processing, for which dealing with the long tail distribution of data can be particularly challenging. Large corpora are available for high resource languages like English while limited data are collected for low resource languages such as Swahili and Urdu. Even within the same language, there are many corner cases or expressions that are valid but extremely rare. As a result, $n$, the number of samples required by the most data-efficient learner, can be huge for samples of low resource languages or even uncommon cases in common languages. Lethal Dose Conjecture suggests that they are also more vulnerable to data poisoning.
> > >
> > > **Please let us know if you want any further explanations! Thanks again for all your feedback~**

---

> > > > ### Comment · Reviewer_7KLu · 2022-08-08
> > > > **Discussion**
> > > >
> > > > Thanks for the running examples --- my questions are all addressed. I'm looking forward to seeing more follow-up work in this field.

---

### Official Review · Reviewer_PsFt · 2022-07-09

**Rating:** 7
**Confidence:** 3
**Soundness:** 4 excellent
**Presentation:** 4 excellent
**Contribution:** 4 excellent

**Summary:**

This paper introduces the Lethal Dose Conjecture (LDC) which speculates on the
minimum amount of bad or poisoned data samples that a learning algorithm can
tolerate (or an attacker can manipulate) before its ability to predict breaks
down. They conjecture that this number is bounded by Θ(N / n) (using "Big
Theta" asymptotic notation) for datasets of size N and problems where a minimum
of n correctly labeled examples are needed. They seek to prove this in three
restricted settings and provide evidence suggesting it may be true in general.

They relate this conjecture to two previously published ensembling techniques
(DFA and FA) that provide robustness when learning on a "poisoned" dataset.
They show that if LDC is true, then DFA and FA are the best mitigations against
poisoned data in the limit as the dataset size grows.

A recurring theme is that data efficiency (the ability to fit a pattern with
fewer examples) is linked to how well a learning algorithm can withstand
poisoning. Experiments use data augmentation to increase the data efficiency of
network-in-network learners on CIFAR-10 and GTSRB.


**Questions:**

Please list up and carefully describe any questions and suggestions for the
authors. Think of the things where a response from the author can change your
opinion, clarify a confusion or address a limitation. This can be very
important for a productive rebuttal and discussion phase with the authors.

There are several nitpicks and typos.

Line 33: I find the "by doing nothing" somewhat unclear.

The use of the term "certified robustness" is used several times, but isn't
defined. Adding that definition in the introduction would be helpful. It looks
like line 79 and 80 are getting at the definition, but I think it would be best
to make the term crystal clear.

I'm fairly sure the term "asymptotically" refers to the growth of the dataset,
but that should be stated.

In Figure 1, you forgot a `\` in front of `mu_1`.

In Section 6, (and in the abstract) the upfront claim is that you are verifying
the conjecture in "Gaussian Classification", but upon further reading it is
more restrictive than that. The Gaussian's have equal co-variance and it seems
that the learning is happening in a balanced setting where each class has
roughly the same number of examples. It isn't clear if the result holds beyond
this case, but the proof seems to specifically address this case. I think this
"balanced equal co-variance" or some other wording to that effect is needed when
introducing this claim (at least in the list of contributions and in the title
of section 6 or the first sentence).

This also raises the question: What happens in an unbalanced setting? How does
that relate to `n` the minimum number of clean examples needed? When discussing
the number `n` there seems to be a presumption that the minimum number of
examples. I'm not sure if this is worth stating in the paper or not.

In the appendix, I did my best to verify the proofs. Nothing stood out as
obviously wrong, but there were places that I think could benefit from further
explanation (at least for me, perhaps its clear to others).

After line 5, (lets label them 5.1 - 5.5) I don't see how to get from 5.1 to
5.2.

On line 8, it would be more formally correct to say ∈ Θ(k) as Θ represents a
set of functions of which log(2 - 2τ)/log(1 - 2/k) is a member.  This comment
applies elsewhere. This is minor as the intention should be generally
understood, but I wanted to point that out.

On line 19, it wasn't clear to me why `E[| T(D_N) - D_N |] = 2N / k`.

I also marked that I didn't understand steps 46.2 and 46.3.

It may be of some value to put some of these proofs or parts of these proofs
into a proof validator and provide that encoding to provide readers like myself
with marginal theoretical ability to feel more confident about their
correctness.  You could codify difficult to encode arguments as assumptions,
which would mean only the truth of the important concepts would need to be
verified.

I was unable to follow lemma 6 and 7. They would benefit from more explanation.
What is the intuition for taking ε -> 0?



**Limitations:**

The authors do not discuss the potential negative impact of this work, but I
think this potential is fringe. In some sense the conjecture could give a
malicious actor a hint on the amount of work they need to do, but it also lets
others know what they need to guard against. I don't think there is any
immediate or zero-day exploit that this enables, so in general I think making
practitioners aware of this (potential) Θ(N / n) relationship is a net social
good.

**Strengths And Weaknesses:**

Please provide a thorough assessment of the strengths and weaknesses of the
paper, touching on each of the following dimensions: originality, quality,
clarity and significance. You can incorporate Markdown and Latex into your
review. See /faq.

The paper is generally well written, well motivated, and provides a good amount
of evidence in support of the claim.

I'm not deeply involved in the related literature on data poisoning, so I can't
comment on how original the work is, but it seems to be a logical (but
non-trivial) next direction given the previous works cited in section 1 and 2.

Apart from having a well-chosen name, the actual conjecture provides a useful
conceptual framework for practitioners to think about data poisoning (or as a
limiting case relating to learning on dirty datasets in general). The N/n
figure makes intuitive sense but nailing it down to exactly that case is
theoretically import.

The code to reproduce the experiments is provided.

I think the biggest weakness is the choice of datasets and experiments.  I
think what is presented is sufficient given the quality of the theoretical
analysis, but I would have liked to see the claim tested against a large
(image-net sized) dataset with a large model, even if the result is: it doesn't
follow the theoretically predicted trend, but that could just be suggesting we
haven't found efficient enough learners.

---

> ### Author Response · Authors · 2022-08-02
> **Response to Reviewer PsFt: part 1**
>
> Firstly we want to thank you for reviewing and especially checking our proofs. We really appreciate feedback regarding how our proofs are presented! Our responses to your questions are as follows.
>
> **(1)** Weakness: ‘I think the biggest weakness is the choice of datasets and experiments. I think what is presented is sufficient given the quality of the theoretical analysis, but I would have liked to see the claim tested against a large (image-net sized) dataset with a large model, even if the result is: it doesn't follow the theoretically predicted trend, but that could just be suggesting we haven't found efficient enough learners.’
>
> **Answer:**
> Thank you for the suggestion. Our code (included in Supplementary Material) is built from the public, official implementation of DPA (https://github.com/alevine0/DPA), which is why we evaluate empirically on datasets that the DPA work  uses, i.e. CIFAR-10 and GTSRB. Notably, our theoretical analysis in Section 7.1 shows that the design of DPA implies the same scaling rule as the one in Lethal Dose Conjecture, meaning that it will follow the same rule on ImageNet as long as we are still using DPA.
> Nevertheless, we do agree evaluation on ImageNet can be interesting to our community, despite the fact that we may not have enough time to do so right away.
>
>
> **(2)** Question: ‘Line 33: I find the "by doing nothing" somewhat unclear.’
>
> **Answer:**
> Thanks for pointing this out. It means that the dataset is clean and not poisoned. For example, if the goal of the adversary is to mislead the model into predicting ‘cat’ for a ‘dog’ image, the assumption by Mahloujifar et al.[21] says that when training on a clean dataset, the classification algorithm must predict the wrong label ‘cat’ with a non-negligible probability (i.e. the probability is at least 1/poly(N), where N is the number of samples).
> This is in fact a very strong assumption, as we can almost always construct classifiers with negligible errors (i.e. the probability reduces exponentially with the number of samples N).
> For instance, considering a binary classification, if a classifier f has an accuracy of 50.1% using 1000 samples, then when you have N samples, you can divide them into groups of 1000 samples each and use the majority votes of f trained on individual groups as final predictions (similar to DPA). With the Chernoff bound, we know this construction gives an algorithm with exponentially decaying (respect to the number of samples N) error rate, which is negligible and does not fit in their assumption.
>
>
> **(3)** Question: For the claims and assumptions regarding ‘Gaussian Classification’.
>
> **Answer:**
> Indeed we assume an isotropic Gaussian for each class for simplicity of the proofs, mostly because there are no simple forms (or none that we know of) for the total variation distance between two arbitrary Gaussian distributions. **However**, our results generalize  to unbalanced settings, which we will discuss in our response to the next question~
> Meanwhile, we used ‘Isotropic Gaussian Classification’ as the new name for the ‘Gaussian Classification’ setting in the updated version of our paper.
>
> **(4)** Question: ‘What happens in an unbalanced setting?’
>
> **Answer:**
> First let us see how Lethal Dose Conjecture applies to an unbalanced setting.
> Notably, Lethal Dose Conjecture is a **pointwise** statement rather than a distributional one: For a (test) sample $(x_0, y_0)$, we uncover the relationship between the difficulty of learning how to predict accurately **on** $\mathbf{x_0}$ and the portion of poisoned samples that one can possibly tolerate while ensuring accuracy **on** $\mathbf{x_0}$.
>
> This is consistent with our intuitions as empirically we always observe that samples are not equally difficult, and naturally they are not equally vulnerable under poisoning attacks. **When the training distribution is unbalanced**, some $x_0$ may become easier as we may need less clean samples drawn from that distribution to learn how to predict $x_0$, and therefore we may tolerate more poisoned samples while ensuring accuracy on $x_0$; Some $x_0$ may become harder and therefore more vulnerable under data poisoning attacks.
>
> **As for the ‘Gaussian Classification’**, an unbalanced setting will not be **geometrically** as interpretable as the one we present, because now the maximum likelihood prediction $y_0$ does not directly correspond to the closest center of Gaussian distributions. Our proofs, however, generalize to the unbalanced setting because we can still compute how far a class center needs to be shifted for the poisoning attack to succeed and how large the corresponding total variation distances are.

---

> > ### Author Response · Authors · 2022-08-02
> > **Response to Reviewer PsFt: part 2**
> >
> > **(5)** Question: ‘After line 5, (lets label them 5.1 - 5.5) I don't see how to get from 5.1 to 5.2.’
> >
> > **Answer:**
> > From line 5.1 to 5.2 in the appendix, what we do is to divide the probability into two cases and bound them separately. Recall the definition of $E$ in line 6 where $E$ denotes the event that all other $k-1$ labels appear in the training set $D_n$.
> > Case 1 is when $E$ happens, where we simply upper bound the probability that $f_{D_n}(x_0)=y_0$ by 1.
> > Case 2 is when $E$ does not happen, meaning that there is some $y_1 \neq y_0$ that does not appear in $D_n$. By Definition 1, we have $Pr [f_{D_n}(x_0) = y_0] = Pr[f_{T_{y_0\leftrightarrow y_1}(D_n)}(x_0) = y_1] = Pr[f_{D_n}(x_0) = y_1]$ thus $Pr [f_{D_n}(x_0) = y_0]\leq \frac{1}{2}$.
> >
> > We have added further explanations about this step in the revised draft.
> >
> >
> > **(6)** Question: ‘On line 19, it wasn't clear to me why E[| T(D_N) - D_N |] = 2N / k.’
> >
> > **Answer:**
> > Note that here $T=T_{y_0\leftrightarrow y_1}$, which is a transform swapping labels $y_0$ and $y_1$ in the training set. Thus $E[|T(D_N) - D_N|]$ is in fact the expected number of samples with a label of $y_0$ or $y_1$, which is $\frac{2N}{k}$. We have added further explanations about this step in the revised draft.
> >
> > **(7)** Question: ‘I also marked that I didn't understand steps 46.2 and 46.3.’
> >
> > **Answer:**
> >
> > For 46.2: When $u_i = v_i$ for all $ i $, we have $  f( \\{u_i\\})  - f( \\{v_i\\}) = 0$ ; When there exists $u_i\neq v_i$ for some $i$, we have $f( \\{u_i\\})  - f( \\{v_i\\}) \leq 1$ because the output of $f$ is $\\{0,1\\}$.
> >
> > For 46.3, we use the union bound. The probability that for at least one $i$ we have $u_i\neq v_i$ is upper bounded by the sum of probability that $u_i\neq v_i$ for all $i$.
> >
> > We have added further explanations about these steps in the revised draft.
> >
> >
> > **(8)** Question: ‘I was unable to follow lemma 6 and 7. They would benefit from more explanation. What is the intuition for taking ε -> 0?’
> >
> > **Answer:**
> > Intuitively, what we do is to construct a second, perfectly legit distribution that is not far from the original one (measured with the total variation distance), so that any classifier must either fail on the original one or fail on the one we construct.
> >
> > If it fails on the original one, the adversary achieves its goal even without poisoning the training set. If it fails on the one we construct, the adversary can still succeed by poisoning only a limited fraction of the training set because the distribution we construct is close to the original one (measured with total variation distance).
> >
> > Regarding the intuition for taking $\epsilon \to 0$: When $\epsilon$ is actually 0, the distributions we construct for different classes will be ‘symmetric’ to $x_0$, meaning that there will be a tie in defining the maximum likelihood prediction. For any $\epsilon >0$, the tie will be broken. By letting $\epsilon \to 0$, we find the tightest bound of the number of poisoned samples needed from our construction.
> >
> > We have added further explanations about these proofs in the revised draft.
> >
> > **Again, thank you for checking our proofs. Please consider raising the score if we do help address your concerns!**

---

### Official Review · Reviewer_2ZuC · 2022-07-12

**Rating:** 6
**Confidence:** 4
**Soundness:** 2 fair
**Presentation:** 3 good
**Contribution:** 3 good

**Summary:**

This work proposes the *Lethal Dose Conjecture* regarding learnability under training-set attacks -- in particular on the amount of data needed by an attacked (sub)model.  The authors demonstrate their conjecture on simple model classes (e.g., instance based-learners).  They also discuss approaches to approve the performance of existing certified defenses generally.

**Questions:**

In Sec. 7.2, the argument is made that since DPA is asymptotically optimal, then improving robustness "reduces developing stronger defenses to finding more data-efficient learners."
* I understand the origin of this claim, but it seems overbroad.  Could it not also be argued that a better/alternative approach is better ways to determine $\ell_0$ robustness of the individual models beyond the assumption that a single insertion/deletion can arbitrarily change the prediction?
* Under your claim the number of models may need to grow to $n$ which affects inference time efficiency. If there was some way -- say even an oracle -- to quantify the intrinsic robustness of each submodel, would that not be similarly as good?  If not why?

Assuming the conjecture is true, in what case should an ensemble-based method like DPA be used?  For simplicity of discussion, consider DPA (not FA) with disjoint partitions.
* By partitioning the training set, an attacker with perfect knowledge can ignore $\frac{k}{2}$ of the submodels or in other words about $\frac{1}{2}$ of the training data.  If only $n$ instances are needed to learn the distribution, does this not reduce the best case bound by (about) half specifically if we can certify the robustness of an individual submodel above 1 (as possible with KNN)?
     * In other words, is partitioning merely a way to dilute the effect of "overwhelmingly lethal" instances?
     * I think this distinction and discussion is more and to some degree glossed over in Section 7.
* Granted the claim in your paper is that they are only optimal up to a constant factor and what I describe is a (constant) factor of 2.
* I think clarity on this point is particularly important for me and would improve the paper.

How do your "baseline" experimental results correspond to the published [DPA implementation](https://github.com/alevine0/DPA)?  Does it take out the data augmentation already in the implementation and compare against that?

**Limitations:**

I believe the finding about data efficiency affecting DPA/FA's performance is an obvious one and already known.  Described very briefly, an ensemble's performance sits on top of the performance of its submodels.  If the submodels are made better, the ensemble will improve.
* For example, without the base augmentation already in [DPA's implementation](https://github.com/alevine0/DPA/blob/ce25c36721f7529350700b071b9a5e1281c31009/FeatureLearningRotNet/dataloader.py#L97), DPA performs much worse.
* I am open to be convinced if others think this finding is less obvious.

Gao et al. [9] study and provides bounds under instance targeted poisoning attack. The relationship of your work to Gao et al. [9] to your work merits a longer discussion than the partial sentence at the end of Section 2.
* As more general feedback, I appreciate when in related work the authors include in the text the author(s)'s names with the citation number. I missed that you even noted Gao et al. the first time I read it because it only had the citation number.

**Strengths And Weaknesses:**

Off the top, there are definitely things I like about this paper; at the same time, I have concerns.  My score is currently set to borderline, and the score changed multiple times over the course of thinking about the paper and writing the review. It is my *a priori* expectation this score will change based on discussions with the authors and other reviewers.

The paper is overall very well-written.  The arguments are laid out clearly.  It uses the right amount of notation to maximize understanding.  Overall, I enjoyed reading it.  I recognize the obvious level of care and thought in the writing.

I also recognize that there is implicitly some level of courage in writing a conjecture paper and that reviewers should be understanding that even a wrong conjecture can advance science.  Nonetheless, I think the bar for such papers should be quite high, and most conjecture papers may be better suited as a preprint unless especially incisive.  I am not convinced this paper meets that latter bar in particular given related work.

I appreciate the demonstration of the conjecture on simple model classes (e.g., bijective learner, instance-based memorized learner).  There are special properties to those learners that make me concerned of if/how readily the conjecture generalizes.  I understand that the argument is made that the conjecture applies to the two extremes of learnability -- hardest and easiest -- so in theory it applies elsewhere.  I would have hoped had the conjecture been demonstrated on even a simple parametric model class (e.g., a linear model); I also understand that this easier said than achieved.  Overall, I am concerned the claims may be overstated.

I thought the point made in Section 7.3 "*If assuming the conjecture is true, when some defense against data poisoning is clearly more robust than DPA (even in restricted cases), there should be a more data-efficient learner.*" was well-stated and effective.  I think the paper would be improved if that point was made more clearly earlier.
*  As a note, I think there may be a typo at the beginning of that sentence.

I appreciate that the authors consider both DPA and FA.  I think the exposition would be clearer focusing exclusively on DPA and having a statement in the related work about how the arguments made about DPA generalize to FA.  DPA is the older method and the gains from FA are often quite marginal and focused mostly in the tail of the certification bound curve (Wang et al. 2022 Table 1).
* Granted FA is newer and can lead to improvements at the expense of increased training and inference costs.  I do not think repeatedly mentioning FA here achieves much beyond making FA better known.

---

> ### Author Response · Authors · 2022-08-02
> **Response to Reviewer 2ZuC: part 1**
>
> Thank you for the review! We especially appreciate the insights you shared--that is the spirit of openreview! Now we will answer your questions and address your concerns.
>
> **(1)** Concern: ‘I understand that the argument is made that the conjecture applies to the two extremes of learnability -- hardest and easiest -- so in theory it applies elsewhere. I would have hoped had the conjecture been demonstrated on even a simple parametric model class (e.g., a linear model); I also understand that this easier said than achieved. Overall, I am concerned the claims may be overstated.’
>
> **Answer:**
> We agree that the claims may be overstated **if** the two cases in Section 4.2 were the only theoretical supports provided, but they are **NOT**. While these two are more intuitive, stronger theoretical supports are provided in Section 5 and Section 6.
>
> In section 5, we provide a ‘distribution discrimination’ view, showing that the very same scaling rule of Lethal Dose Conjecture applies when one wants to discriminate any distributions.
>
> In section 6, we prove Lethal Dose Conjecture for classification, assuming data from each class follow an isotropic Gaussian distribution.
>
> It is the union of the analysis in Section 4.2, Section 5 and Section 6 providing evidence  that the scaling rule in Lethal Dose Conjecture is no coincidence.
>
>
> **(2)** Question: ‘In Sec. 7.2, the argument is made that since DPA is asymptotically optimal, then improving robustness "reduces developing stronger defenses to finding more data-efficient learners." I understand the origin of this claim, but it seems overbroad. Could it not also be argued that a better/alternative approach is better ways to determine $\ell_0$
>  robustness of the individual models beyond the assumption that a single insertion/deletion can arbitrarily change the prediction?’
>
> **Answer:**
> Good question! The rationale behind that argument is that we want to simplify the defense problem through reduction. In particular, to defend against data poisoning, we are trying to design algorithms/models with $\ell_0$ robustness (with respect to the training set) overall. Here, Lethal Dose Conjecture implies that DPA is a nearly optimal reduction from designing $\ell_0$ robust models to designing data-efficient models, **simplifying** a problem with robustness requirements to a problem with none.
>
> This is desirable as now we can focus on a simpler task. Meanwhile, in formulations, making base models $\ell_0$ robust is not easier than making the whole model $\ell_0$ robust.
>
> Here is another way of looking at this: When the base models are already robust against data poisoning, it implies that one can also increase robustness by using more base models with less training data each. In fact, in some sense, an example of this is presented in Section 7.3 of our paper, where we show that a base learner for DPA can be derived from nearest neighbor, an approach with intrinsic robustness. DPA using the derived base learner offers similar robustness as the nearest neighbor method.
>
>
>
>
> **(3)** Question: ‘Under your claim the number of models may need to grow to n which affects inference time efficiency. If there was some way -- say even an oracle -- to quantify the intrinsic robustness of each submodel, would that not be similarly as good? If not why?’
>
> **Answer:**
> Yes, it is totally possible that a method with intrinsic robustness may be as robust as DPA (using the most data-efficient learners) while offering a faster inference. We believe improving inference time can also be a valuable direction for future research. Our conjecture focuses on the extremes of robustness but not inference time.
>
>
> **(4)** Question: ‘In other words, is partitioning merely a way to dilute the effect of "overwhelmingly lethal" instances?’
>
> **Answer:**
> In some sense, yes. The intuition behind DPA is no mystery and it is fair to say that it is some sort of dilution. What is impressive and non-trivial about the Lethal Dose Conjecture is that it implies that such simple dilution is surprisingly good and, as shown in the paper in several cases, is nearly optimal.
>
>
> **(5)** Question: ‘How do your "baseline" experimental results correspond to the published DPA implementation? Does it take out the data augmentation already in the implementation and compare against that?’
>
> **Answer:**
> The baseline results are consistent with the published DPA implementation. DPA_baseline uses the very **same** augmentations and hyperparameters as the published DPA implementation and the results in our Figure 2 matches the corresponding settings reported in the original paper of DPA. We do **not** take out the augmentation already in the implementation. It is surprising but one can indeed double or triple the reported robustness of one of SOTAs! This is because our community has not put much effort into improving base learners and the potential from more data-efficient base learners remains undiscovered.

---

> > ### Author Response · Authors · 2022-08-02
> > **Response to Reviewer 2ZuC: part 2**
> >
> > **(6)** Limitation: ‘I believe the finding about data efficiency affecting DPA/FA's performance is an obvious one and already known.’
> >
> > **Answer:**
> > This is only partially true.
> > It is indeed obvious that more data-efficient base learners can improve DPA/FA’s performance. **However**, previously we knew only that this is one way towards our destination, now we know that this can be one of the optimal ways!
> >
> > In addition, this offers a new motivation for advancing machine learning with small training sets and/or few-shot learning (even in domains where data is abundant), where the techniques differ greatly from learning with large datasets. We hope the conjecture will also facilitate the advancement of relevant fields.
> >
> >
> > **(7)** Limitation: ‘ The relationship of your work to Gao et al. [9] to your work merits a longer discussion than the partial sentence at the end of Section 2. As more general feedback, I appreciate when in related work the authors include in the text the author(s)'s names with the citation number.’
> >
> > **Answer:**
> >
> > Thanks for the suggestion regarding citation formats! We updated it in the revised version of the paper. Due to page limits, we included further discussion with regards to the Gao et al. [9] in Appendix in the updated version and will include it in the main paper for the camera-ready version which allows an additional content page.
> >
> > In Gao et al. [9], authors make a very creative step towards understanding how the budget of data poisoning attacks and the size of the training set interact and affect whether we can defend the attacks.
> >
> > However, the main results of Gao et al. [9] can in fact be implied by Lethal Dose Conjecture. In this sense, the conjecture is stronger and more general.
> >
> > For example: Let $N$ be the size of the training set and $m$ be the number of poisoned samples. Lethal Dose Conjecture implies that the 'Lethal Dose' (the threshold for when poisoning attacks can be too strong to be defended) is $m/N \approx \Omega(1/n)$, where $n$ is the number of samples needed by the most data-efficient learners to achieve accurate predictions. Meanwhile, Theorem 3.2 and 3.3 of Gao et al. [9] only suggest when $m=o(N)$, i.e. $m/N \to 0$, the poisoning attacks are defendable.
> >
> >
> > **Thank you for reading! Please consider raising the score if we do help addressing your concerns~**

---

> > > ### Comment · Reviewer_2ZuC · 2022-08-08
> > > **Updated Score**
> > >
> > > I appreciate your detailed and thoughtful response not only to me but to the other reviewers.  While I still have some concerns about some of the claims, I think the paper's contribution is durable even if the conjecture does not always hold and insights into the task of improving (certified) robustness.
> > >
> > > I have increased my score and believe this paper merits inclusion at NeurIPS.

---

> > > > ### Author Response · Authors · 2022-08-08
> > > > **Thank you!**
> > > >
> > > > Thank you for letting us know!
> > > >
> > > > We appreciate all your comments, questions, and services!
> > > >
> > > > Meanwhile, just to let you know, you are welcome for any follow-up questions~

---

### Author Response · Authors · 2022-08-08
**To Reviewers: A Reminder**

Now the reviewer-author discussion phase will end in a couple of days,
we want to thank you again for the valuable reviews and services!

Could you please let us know if our responses address your concerns or do you have follow-up questions?

We truly appreciate and value any feedback from you!

---

### Meta-Review · Area_Chair_PA5m · 2022-08-22

**Recommendation:** Accept
**Confidence:** Certain

**Metareview:**

The reviewers agree that this work proposes an interesting conjecture which is likely to inspire further research.
Congrats!

During the discussion period the following two points were raised by the reviewers:

- The paper should emphasize more strongly that this is just a conjecture (I at minimum have doubts have how well it generalizes). They should make clearer the alternate hypotheses/explanations in the main paper and discuss them.
- Nascent researchers look at archived OpenReview discussions and may adopt similar styles as successful authors. I do not think the authors' approach of ending every post with a plea to increase the score is an appropriate or healthy style for peer review. I further advocate it definitely should not be emulated/copied. If reviewers believe scores should be raised (as I did), they should be trusted to do so without pressure from authors.



**Award:**

No

---

### Decision · Program_Chairs · 2022-09-14

Accept